# Natural Compounds That Enhance Motor Function in a Mouse Model of Muscle Fatigue

**DOI:** 10.3390/biomedicines10123073

**Published:** 2022-11-29

**Authors:** Shuichi Shibuya, Kenji Watanabe, Daiki Sakuraba, Takuya Abe, Takahiko Shimizu

**Affiliations:** 1Aging Stress Response Research Project Team, National Center for Geriatrics and Gerontology, Obu 474-851, Aichi, Japan; 2Zenyaku Hanbai Co., Ltd., Toshima-ku 170-0013, Tokyo, Japan

**Keywords:** running activity, reactive oxygen species, muscle function

## Abstract

Musculoskeletal disease can be a serious condition associated with aging that may lead to fractures and a bedridden state due to decreased motor function. In addition to exercise training to increase muscle mass, increasing muscle function with the intake of functional foods is an effective treatment strategy for musculoskeletal disease. Muscle-specific SOD2-deficient mice (muscle-*Sod2*^-/-^) show a severe disturbance in exercise in association with increased mitochondrial reactive oxygen species, as well as mitochondrial dysfunction and muscle damage. In the present study, to develop a therapeutic strategy for musculoskeletal disease, we searched for substances that enhanced motor function among functional compounds by in vivo screening using muscle-*Sod2*^-/-^ mice as a muscle fatigue model. We administered 96 compounds, including antioxidants, to muscle-*Sod2*^-/-^ mice and assessed their effects on treadmill performance. Among the administered compounds, gossypin, genistein, kaempferol, taxifolin, fumaric acid, β-hydroxy-β-methylbutyrate Ca, and astaxanthin, which are dietary functional food factors, increased forced running time in muscle-*Sod2*^-/-^ mice. In addition, troglitazone, tempol, trolox, and MnTE-2-PyP, which are antioxidants, also significantly increased the running ability of muscle-*Sod2*^-/-^ mice. These results suggest that the intake of functional foods with antioxidant activity can improve motor function. Muscle-*Sod2*^-/-^ mice, as a muscle fatigue model, are suitable for the in vivo screening of functional substances that promote improvements in exercise and muscle performance.

## 1. Introduction

Reactive oxygen species (ROS) cause tissue dysfunction by oxidizing the proteins, lipids, and nucleic acids that constitute living organisms. Oxidative damage markers, such as carbonylated protein, lipid peroxides, and oxidized nucleic acids, increase the aging process and are associated with the development of various diseases, suggesting a strong correlation between ROS and various tissue pathologies, including skeletal muscle pathologies. Actually, oxidative damage markers accumulate in aged rat skeletal muscle [1]. In humans, mitochondria derived from aged skeletal muscle also show an increase in the oxidized nucleic acid marker 8-OHdG, which causes respiratory dysfunction accompanied by a decrease in mitochondrial DNA [2]. The accumulation of ROS in skeletal muscle also contributes to the pathogenesis of muscular dystrophy, which causes progressive skeletal muscle wasting and degeneration [3]. These reports indicate that the redox imbalance caused by increased ROS induces muscle hypofunction.

Mammals have various antioxidant systems that protect tissues and cells from ROS. Superoxide dismutase (SOD) is an antioxidant enzyme that converts superoxide to hydrogen peroxide, which is further detoxified to water and O_2_ by catalase and GPx enzymes. SOD2 is constitutively and ubiquitously expressed in the mitochondria to regulate the redox balance in the cells of tissues. The loss of SOD2 induces mitochondrial redox imbalance by increasing the generation of superoxide, resulting in mitochondrial dysfunction in cells and several tissues, including the brain, heart, muscle, bone, and cartilage [4,5,6,7]. Muscle-specific SOD2-deficient (muscle-*Sod2*^-/-^) mice show a severe disturbance during exercise activity in association with increased oxidative damage and reduced ATP content [4]. Meanwhile, the administration of the antioxidant EUK8, which has superoxide dismutase and catalase activity, was shown to improve the motor activity of muscle-*Sod2*^-/-^ mice [4]. Muscle-*Sod2*^-/-^ mice can be used as a model of muscle fatigue to evaluate the effects of various compounds, including antioxidants, on the improvement of the motor function.

In this study, we evaluated the improvement effects of polyphenols, vitamins, amino acids, and other chemicals by an in vivo screening of the forced running ability of muscle-*Sod2*^-/-^ mice.

## 2. Materials and Methods

### 2.1. Animals and Genotyping

The generation of muscle-*Sod2*^-/-^ mice was described previously [4]. In our previous report, a 401-bp DNA fragment corresponding to the deleted allele was specifically amplified by PCR from the skeletal muscle of the muscle-*Sod2*^-/-^ mice, whereas no fragments were amplified from other tissues in the muscle-*Sod2*^-/-^ mice [4]. Western blot analyses also showed a specific loss of SOD2 expression in skeletal muscles, such as the tibialis anterior, gastrocnemius, soleus, and quadriceps, of muscle-*Sod2*^-/-^ mice [4]. The muscle-*Sod2*^-/-^ mice, which were all male (age: 6–9 months), were maintained under a 12 h light/12 h dark cycle with ad libitum access to water and chow. All genotyping of the HSA-Cre transgene and the SOD2^lox/lox^ mice was performed by PCR using genomic DNA isolated from the tail tip. The primers used for identifying carriers of the HSA-Cre transgene were 5′-AAATACTCTGAGTCCAAACCGGGCCCC-3′ and 5′-CAGTGCGTTCGAACGCTAGAGCCTGTT-3′, while those used for genotyping SOD2 lox were 5′-TTAGGGCTCAGGTTTGTCCAGAA-3′, 5′-CGAGGGGCATCTAGTGGAGAA-3′, and 5′-AGCTTGGCTGGACGTAA-3′. The mice were maintained and studied according to protocols approved by the Tokyo Metropolitan Institute of Gerontology and the National Center for Geriatrics and Gerontology.

### 2.2. Substances

We selected the substances administered in this study from among the available substances, focusing on those with antioxidant, anti-inflammatory, and mitochondrial biosynthetic effects. The administered substances were all purchased from or provided by the following suppliers: Dojindo Laboratories (Kumamoto, Japan), FUJIFILM Wako Pure Chemical Corporation (Osaka, Japan), KOBAYASHI PERFUMERY CO., LTD. (Tokyo, Japan), Kyoto University (Kyoto, Japan), LKT Laboratories (St. Paul, MN, USA), and Merck (Darmstadt, Germany). Detailed information on the administered substances is provided in Table 1 and Table 2.

### 2.3. Administration

Many substances were administered by intraperitoneal (I. P.) injection, which is a simple procedure that can be performed with small amounts of expensive substances. The compounds used for I. P. administration were diluted in PBS and injected once into muscle-*Sod2*^-/-^ mice at the dosages indicated in Table 1 and Table 2. Treadmill exercise was performed 24 h after administration. Some compounds used for oral administration were diluted in distilled water and administered once per day to muscle-*Sod2*^-/-^ mice at the dosages and frequencies indicated in Table 1. A schematic illustration of the administration and treadmill experiments is shown in Figure 1. Mice that completed the treadmill exercise were used again for the evaluation of other compounds after a 2-week clearance period. The dosage of each substance was determined with reference to previous reports using experimental animals. To assess the safety, we administered each substance to wild-type mice and confirmed that no deaths or abnormalities occurred.

### 2.4. Treadmill Protocol

A treadmill apparatus (MK-680S/OP; Muromachi Kikai, Tokyo, Japan) was used to determine the endurance capacity for running. All mice were accustomed to the treadmill before the experiment, and the running test was carried out at 12 m/min with a 0° slope. An electrode at the back of the treadmill was activated to prevent mice from stopping naturally.

### 2.5. Statistical Analysis

The minimum number of mice in each experimental group was set at four for statistical analyses. Student’s *t*-test was used to compare the running times before and after treatment. *p* values of <0.05 were considered to indicate statistical significance. All data are expressed as the mean ± standard deviation (SD).

## 3. Results

In order to evaluate the improvement effect on motor function, we administered various compounds intraperitoneally or orally to muscle-*Sod2*^-/-^ mice, and evaluated the changes in treadmill running time before and after these administrations. As evaluation compounds, we selected polyphenols, vitamins and their metabolites, carotenoids, amino acids, other phytochemicals, and other compounds with a focus on substances with antioxidant, anti-inflammatory, and mitochondrial biosynthetic effects (Table 1 and Table 2). In our previous report, we demonstrated that wild-type mice could complete a treadmill run for 2 h, whereas muscle-*Sod2*^-/-^ mice ran for less than 10 min [4]. Each experimental group of muscle-*Sod2*^-/-^ mice in this study averaged about 7–23 min of running time before administration (data not shown), indicating exercise intolerance similar to our previous reports. When the forced running time before and after administration was compared, we found 11 compounds that significantly increased running time. When the 11 compounds that increased the running time of muscle-*Sod2*^-/-^ mice were divided into categories, they contained 7 dietary functional foods (gossyppin, genistein, kaempferol, taxifolin, fumaric acid, astaxanthin, and β-hydroxy-β-methylbutyrate (HMB) Ca) (Figure 2). The antioxidants troglitazone, tempol, trolox, and MnTE-2-PyP also improved the forced running time of muscle-*Sod2*^-/-^ mice (Figure 3). In contrast, the intraperitoneal administration of citric acid, phosphocreatine, and nicotinamide significantly reduced the running time of muscle-*Sod2*^-/-^ mice (Figure 4). These results indicate that muscle-*Sod2*^-/-^ mice can be used as an exercise intolerance model for the in vivo screening of compounds that enhance physical function.

## 4. Discussion

In vivo screening identified 11 compounds that improved the physical function of muscle-*Sod2*^-/-^ mice. Previously, we demonstrated that muscle-*Sod2*^-/-^ mice exhibit exercise intolerance due to mitochondrial dysfunction caused by the accumulation of ROS in skeletal muscle and that the antioxidant EUK8 improves their running ability [4]. In this study, 10 of the 11 compounds that improved motor function were antioxidants, including polyphenols. Genistein (an isoflavone) and astaxanthin (a carotenoid classified as a xanthophyll) are major antioxidants. The administration of tempol, which has SOD activity, decreased oxidative markers along with the regulation of antioxidant enzyme expression in the skeletal muscle of a mouse model of Duchenne muscular dystrophy [8]. These results suggest that the reduction in oxidative damage in SOD2-deficient skeletal muscle is the main mechanism underlying the enhancement in motor function. Gossypin, a flavone isolated from *Hibiscus vitifolious*, has strong antioxidant activity in scavenging superoxide, hydroxyl radicals, DPPH radicals, and NO [9]. In addition, gossypin was also shown to have anti-inflammatory effects via the inhibition of NF-κB and its regulatory gene expression in in vitro studies using inflammatory stimuli and carcinogens [10]. Kaempferol is a flavonol (a type of flavonoid) that is abundantly contained in kale and broccoli. Kaempferol has ROS activity, scavenging superoxide, hydroxyl radical, and peroxynitrite [11,12,13]. In rat studies, the oral administration of the glycoside kaempferol 3-neohesperidoside increased the accumulation of glycogen by upregulating the muscle glucose uptake via the phosphoinositide 3-kinase and protein kinase C pathways [14], suggesting an additive effect—involving antioxidant activity and other mechanisms—on muscle function. Further analyses are needed to investigate the correlation between the enhancement in physical function and the level of oxidative damage in skeletal muscle in greater detail. The pharmacological effects on the motor function of substances may be associated with the pharmacokinetics after administration. In rats, astaxanthin accumulated in skeletal muscle at 24 h after ingestion, suggesting a direct action on skeletal muscle [15]. Human studies also suggest that HMB reaches various organs at nine hours after ingestion [16]. For the majority of this study, we analyzed animals at 24 h after I. P. administration, so the pharmacological action of the substances may have been enhanced if analyzed at the time point of peak pharmacokinetics. Indeed, despite having antioxidant activity, some compounds failed to improve motor function in muscle-*Sod2*^-/-^ mice (Table 1 and Table 2). These substances have different pharmacokinetics, e.g., differences in the amount reaching muscle tissues and the stability, and may have their own appropriate concentrations, analysis timings, and administration methods.

Muscle-*Sod2*^-/-^ mice exhibit mitochondrial dysfunction due to reduced complex II activity [4]. Some of the compounds that showed an improvement effect on the motor function of muscle-*Sod2*^-/-^ mice in the present study are reported to enhance mitochondrial function. Taxifolin, a flavonoid contained in coniferous trees, reduced the intracellular ROS level against H_2_O_2_-induced pyroptosis and exhibited cytoprotective effects associated with increased mitochondrial membrane potential in rat heart cells [17]. In an in vitro study, taxifolin also decreased the lipid peroxidation levels in mitochondria derived from the livers of rats, suggesting that taxifolin may also enhance motor function due to the increased mitochondrial function in skeletal muscle [18]. Trolox, a water-soluble analog of vitamin E, which is known as a major antioxidant, induced the improvement in mitochondrial respiration and the suppression in ubiquitin–proteasome/autophagy activity by attenuating oxidative stress. This resulted in the prevention of muscle atrophy in skeletal muscle-specific transforming growth factor-β-activated kinase 1-deficient mice [19]. Fumaric acid, which is a type of dicarboxylic acid, is a constituent of the citric acid cycle and thus contributes to energy production in muscle.

The uptake of glucose into muscles in glycolysis affects the energy production capacity. Troglitazone, a type of thiazolidine with a tocopherol-like structure, is used as a hypoglycemic and anti-inflammatory medicine. In a glucose clamp test using Zucker fatty rats, troglitazone significantly increased the glycogen content in skeletal muscle [20]. These results suggest that the accelerated glucose uptake in muscles improves exercise capacity by increasing energy sources. HMBCa is a functional food material in which calcium is bound to HMB. HMB is a metabolite of leucine, an essential amino acid, which promotes muscle synthesis, inhibits muscle degradation, and acts as a source of components for muscle cell membranes [21]. Although HMB plays an important role in the maintenance of muscle function, the amount of HMB produced in the body is approximately 5% of the leucine intake [22]; thus, the direct ingestion of HMB is an effective method for increasing muscle function. In the present study, we performed the trials with a limited number of mice and did not examine biochemical markers in detail, thereby limiting the interpretation of the results obtained. In order to clarify the mechanisms through which the compounds evaluated in this study enhance motor function, we also need to examine other biochemical markers, such as ROS levels, ATP levels, and mitochondrial respiratory capacity, in detail.

Musculoskeletal disorders are a serious condition associated with aging, increasing the risk of a bedridden state due to decreased motor function. Muscle-*Sod2*^-/-^ mice are very useful as a muscle fatigue model for the in vivo screening of reagents that enhance motor function. Utilizing this mouse model of muscle fatigue will contribute to the development of materials that can be applied to locomotive syndrome and frailty.

## Figures and Tables

**Figure 1 biomedicines-10-03073-f001:**
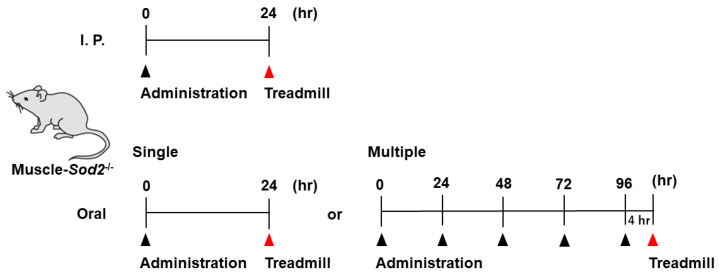
Schematic illustration of the administration and treadmill experiments. We administrated substances by I. P. injection (top) or orally (bottom) to the muscle-*Sod2*^-/-^ mice, followed by treadmill running.

**Figure 2 biomedicines-10-03073-f002:**
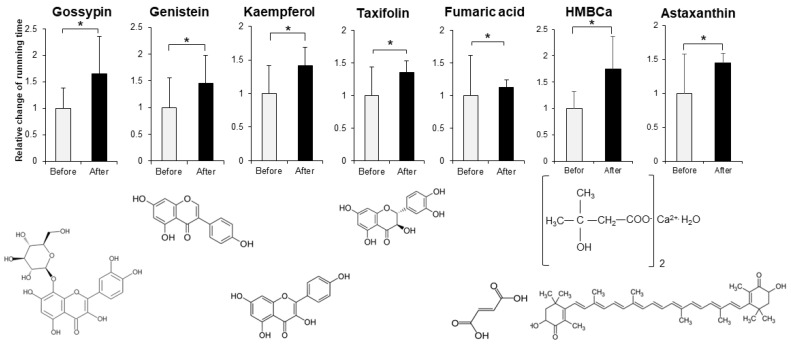
Functional factors that enhance the running activity of muscle-*Sod2*^-/-^ mice. Positive effects of functional factors on the treadmill task performance of muscle-*Sod2*^-/-^ mice (top) and structural formulas of each compound (bottom). * *p* < 0.05. Data are shown as the mean ± SD.

**Figure 3 biomedicines-10-03073-f003:**
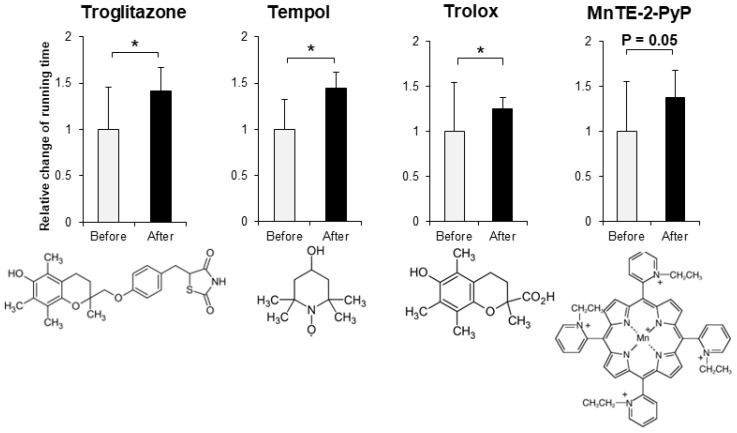
Antioxidants that enhance the running activity of muscle-*Sod2*^-/-^ mice. Positive effects of antioxidants on the treadmill task performance of muscle-*Sod2*^-/-^ mice (top) and structural formulas of each compound (bottom). * *p* < 0.05. Data are shown as the mean ± SD.

**Figure 4 biomedicines-10-03073-f004:**
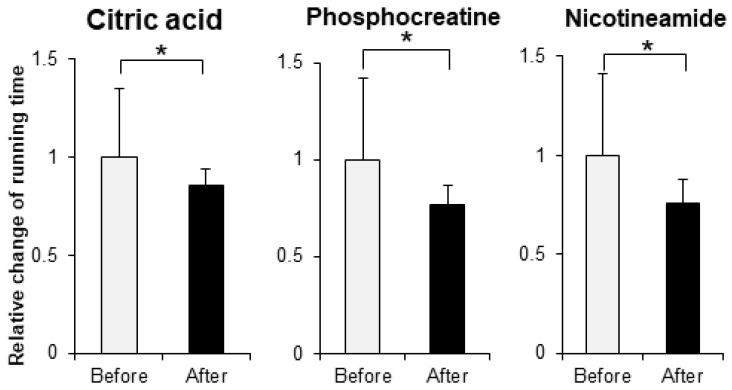
Compounds that reduce the running activity of muscle-*Sod2*^-/-^ mice. Negative effects of some compounds on the treadmill task performance of muscle-*Sod2*^-/-^ mice. * *p* < 0.05. Data are shown as the mean ± SD.

**Table 1 biomedicines-10-03073-t001:** Positive effects of compounds on the running activity of muscle-*Sod2*^-/-^ mice.

Compounds	Supplier	Dosage (mg/kg)	Administration	Relative Change	*t*-Test	n
**Positive effects**						
Acacetin	Merck	50	I. P.	1.26	0.07	5
Aesculin	LKT Laboratories	100	I. P.	1.41	0.36	5
AICAR	Merck	500	I. P.	1.01	0.65	4
Alliin	Merck	10	I. P.	1.29	0.33	5
Allopurinol	Merck	100	I. P.	1.08	0.26	4
Apigenin	Merck	50	I. P.	1.03	0.63	5
Apocynin	Merck	300	I. P.	1.27	0.31	5
Ascorbic acid	Merck	500	I. P.	1.26	0.29	10
Asparagine	Merck	500	I. P.	1.90	0.21	4
Astaxanthin	Merck	500	Oral	1.24	0.04	4
ATP	Merck	2000	Oral	1.07	0.96	4
Auraptene	Kyoto Univ.	30	I. P.	1.24	0.12	4
Baicalein	Merck	500	I. P.	1.44	0.17	5
Biochanin A	Merck	50	I. P.	1.07	0.68	5
Caffeic acid	Merck	50	I. P.	1.33	0.36	5
L-Carnitine	Merck	1690	I. P.	1.25	0.13	14
Chrysin	Merck	250	I. P.	1.17	0.23	4
L-Citrulline	Merck	100	I. P.	1.18	0.47	5
Curcumin	Merck	150	I. P.	1.17	0.10	5
2,7-Dichlorofluorescein	Merck	500	I. P.	1.40	0.35	2
Dipentene	Merck	10	I. P.	1.03	0.78	4
Ebselen	Merck	16	I. P.	1.44	0.19	4
EGCs	Merck	150	Oral	1.14	0.35	5
Fenofibrate	Merck	1600	I. P.	1.25	0.58	5
Ferulic acid	Merck	50	I. P.	1.37	0.11	5
Fisetin	Merck	50	I. P.	1.18	0.62	5
Folic acid	Merck	4	I. P.	1.21	0.14	10
Fumaric acid	Merck	1000	I. P.	1.12	0.04	5
Genistein	Merck	10 and 50	I. P.	1.45	0.01	15
[6]-Gingerol	Merck	10	I. P.	1.15	0.27	4
Glucose	Merck	2000	I. P.	1.08	0.62	5
Glutamine	Merck	1000	I. P.	1.15	0.66	5
Glutamine monohydrate	Merck	1000	I. P.	1.10	0.47	5
Glutathione (GSH-MEE)	Merck	50	I. P.	1.06	0.49	5
Glycine	Merck	150	I. P.	1.24	0.37	5
Gossypin	Merck	500	I. P.	1.65	0.01	14
Histidine	Merck	800	I. P.	1.08	0.31	5
HMBCa	KOBAYASHI PERFUMERY	500 mg/kg × 5 times	Oral	1.44	0.02	10
Kaempferol	Merck	500	I. P.	1.42	0.03	4
Luteolin	FUJIFILM	100	I. P.	1.04	0.91	4
Lysine	Merck	185	I. P.	1.35	0.10	5
Magnolol	Merck	10	I. P.	1.05	0.95	5
Maleic acid	Merck	1000	I. P.	1.25	0.27	5
Methionine	Merck	187	I. P.	1.16	0.13	5
Mevastatin	Merck	500	I. P.	1.18	0.06	4
Mn TE-2-PyP	Merck	10	I. P.	1.38	0.05	5
Morin	Merck	500	I. P.	1.32	0.05	5
Myricetin	Merck	500	I. P.	1.09	0.87	5
L-NAME	Dojindo	10	I. P.	1.86	0.14	5
α-Naphthoflavone	Merck	250	I. P.	1.25	0.26	4
β-Naphthoflavone	Merck	250	I. P.	1.19	0.43	5
Naringin	Merck	500	I. P.	1.12	0.75	5
Nobiletin	Kyoto Univ.	30	I. P.	1.49	0.19	4
1-Octanosanol	Merck	100	I. P.	1.08	0.14	4
Ornithine	Merck	1000	I. P.	1.04	0.57	5
(-)-2-Oxo-4-thiazolidinecarboxylic acid	Merck	50	I. P.	1.26	0.32	5
Prednisolone	Merck	200	I. P.	1.20	0.27	5
Reduced lipoic acid	Merck	10	I. P.	1.25	0.37	5
Rutin	Merck	50	I. P.	1.10	0.57	5
Sodium Pyruvate	Merck	500	I. P.	1.21	0.28	5
Succinic acid	Merck	1000	I. P.	1.07	0.69	5
Taxifolin	Merck	500	I. P.	1.36	0.02	5
Tempol	Merck	250	I. P.	1.36	0.04	5
L-Theanine	Merck	500	I. P.	1.56	0.14	4
Threonine	Merck	800	I. P.	1.13	0.48	5
α-tocopherol acetate	Merck	500	I. P.	1.16	0.18	4
Troglitazone	Merck	10	I. P.	1.41	0.01	5
Trolox	Merck	20	I. P.	1.41	0.01	5
Zn-protoporphyrin IX	Merck	10	I. P.	1.62	0.15	5

**Table 2 biomedicines-10-03073-t002:** Negative effects of compounds on the running activity of muscle-*Sod2*^-/-^ mice.

Compounds	Supplier	Dosage (mg/kg)	Administration	Relative Change	*t*-Test	n
**Negative effects**						
Aloe-emodin	Merck	10	I. P.	0.98	0.97	5
Bee pollen	Merck	100	I. P.	0.88	0.20	5
Bilirubin	Merck	200	I. P.	0.87	0.31	5
Capsaicin	Merck	1	I. P.	0.98	0.55	4
β-carotene	Merck	500	I. P.	0.71	0.63	4
Citric acid	Merck	500	I. P.	0.86	0.02	5
Daidzein	Merck	50	I. P.	0.99	0.78	5
Docosahexaenoic acid	Merck	100	I. P.	0.81	0.21	4
Eicosapentaenoic acid	Merck	100	I. P.	0.86	0.09	5
18β-Glycyrrhetinic acid	Merck	10	I. P.	0.95	0.79	5
Glycyrrhizic acid ammonium salt	Merck	10	I. P.	0.95	0.61	7
Ibuprofen	Merck	75	I. P.	0.87	0.67	4
Indore-3-carbinol	Merck	100	I. P.	0.92	0.69	4
Isosorbide dinitrate	Merck	30	I. P.	0.87	0.19	5
Limonin	Merck	10	I. P.	0.75	0.12	5
Linolenic acid	Merck	100	I. P.	0.87	0.91	5
N-acetyl-L-cysteine	Merck	500	I. P.	0.92	0.53	5
Nicotinamide	Merck	200	I. P.	0.76	0.03	5
7-Nitroindazole	Merck	50	I. P.	0.97	0.51	5
Phosphocreatine	Merck	100	I. P.	0.77	0.01	5
Quercetin	Merck	3.3	I. P.	0.96	0.52	5
Riboflavin	Merck	100	I. P.	0.77	0.12	5
Rosmarinic acid	Merck	50	I. P.	0.99	0.62	5
Sesamin	Merck	5	I. P.	0.88	0.28	5
Ursolic acid	Merck	100	I. P.	0.97	0.33	5
Xanthophyll	Merck	5	I. P.	0.97	0.55	4

## Data Availability

Not applicable.

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
