# Peer review of "Natural Compounds That Enhance Motor Function in a Mouse Model of Muscle Fatigue"

_biomedicines, 2022, doi:10.3390/biomedicines10123073_

Round 1

Reviewer 1 Report

This study investigated the effect of a series compounds on improving the muscle fatigue intolerance in a mouse model of muscle specific SOD2 deficient mice. Treadmill exercise were used before and after IP injection or oral administration of the compounds, for comparing the muscle resistance to fatigue. This study appears to have been rigorously conducted and analyzed, however, there are some major deficits in the experimental design or the data presentation. My few concerns are as follows:

1.     Muscle specific SOD2 deficient mice as the unique mouse model used in this study, the Authors did not provide any genotype validation for this mouse model, i.e. western blotting evidence showing the removal of SOD1? If it is a muscle specific genetic manipulation, a tissue panel western blot should be provided to show that this manipulation uniquely occurred in muscle, not in any other tissues.

2.     In addition to point 1, muscle-Sod2-/- mouse is the only mouse model used in this study, and there is no wildtype control mouse used, so it is really hard to get any sense of the influence of Sod2 knockout on muscle performance. At least, without administration of the compounds, what does it look like of the running time and distance in muscle-Sod2-/- mouse and wildtype mouse?

3.     Lack of biochemical evidence. After the administration, some of the compounds show promising effects in improving muscle functions in muscle-Sod2-/- mouse, any further interpretation on that? For example, as mentioned by the Authors in the introductions, in muscle-Sod2-/- mouse model, the muscle fatigue intolerance could be attributed to impaired mitochondrial function, increased ROS or oxidative stress, damaged or oxidized proteins with elevated oxidative markers. Then if the muscle fatigue intolerance situation was alleviated by the administration of some compounds, was it due to the attenuation of the oxidative stress, or reduced ROS level, or increased mitochondrial function?     

4.     Vulnerable design of the compounds administration. For the compounds delivery, there is only one time administration to the mice for improving their treadmill performance, I highly doubt the outcome of this “single treatment”. Is there any proof to show the pharmacokinetic of the targeted compound, any evidence to show the levels of the compound in the blood and in the muscle following time?

5.     Lack of rationale on picking up so many compounds. There are so many compounds tested in the study, with most of them having no significant effect, and also there is not enough explanation on why picking up these certain compounds. As a reader, I’d rather see fewer compounds with significant effect, and also with the rationale of why choosing these compounds, and why they would be effective in preserving muscle functions in   muscle-Sod2-/- mouse.

6.     The data presented in this study is only the relative change of the running time with arbitrary unit. I have no clue about their exact running time, and the actual running distance. I think the absolute running time and distance would make more sense to the readers, and also will be more comparable to studies by other peers, at least for the running exercise data in wildtype mouse.  

Author Response

Response to Reviewer 1

This study investigated the effect of a series compounds on improving the muscle fatigue intolerance in a mouse model of muscle specific SOD2 deficient mice. Treadmill exercise were used before and after IP injection or oral administration of the compounds, for comparing the muscle resistance to fatigue. This study appears to have been rigorously conducted and analyzed, however, there are some major deficits in the experimental design or the data presentation. My few concerns are as follows:

  1. Muscle specific SOD2 deficient mice as the unique mouse model used in this study, the Authors did not provide any genotype validation for this mouse model, i.e. western blotting evidence showing the removal of SOD1? If it is a muscle specific genetic manipulation, a tissue panel western blot should be provided to show that this manipulation uniquely occurred in muscle, not in any other tissues.

Response: We generated skeletal-muscle-specific SOD2-deficient mice (muscle-Sod2−/−) using the Cre-loxP system under the control of the human skeletal actin (HSA) promoter (Kuwahara et al. Free Radical Biology & Medicine 2010). We previously confirmed the deletion of the allele in skeletal muscle but not in the kidney, liver, heart, or brain in muscle-Sod2−/−. Furthermore, a Western blot analysis revealed that the SOD2 protein was deleted only in skeletal muscles in muscle-Sod2−/−. We have now added a description of the genotype validation of this mouse model to the ‘Materials and Methods’ section.

‘In our previous report, a 401-bp DNA fragment corresponding to the deleted allele was specifically amplified by PCR from the skeletal muscle of the muscle-Sod2-/- mice, whereas no fragments were amplified from other tissues in muscle-Sod2-/- mice [4]. Western blot analyses also showed a specific loss of SOD2 expression in skeletal muscles, such as the tibialis anterior, gastrocnemius, soleus, and quadriceps, of muscle-Sod2-/- mice [4].’

  1. In addition to point 1, muscle-Sod2-/- mouse is the only mouse model used in this study, and there is no wildtype control mouse used, so it is really hard to get any sense of the influence of Sod2 knockout on muscle performance. At least, without administration of the compounds, what does it look like of the running time and distance in muscle-Sod2-/- mouse and wildtype mouse?

Response: In our previous report, we demonstrated that wild-type mice can complete a treadmill run for 2 h, whereas muscle-Sod2−/− mice run for less than 10 minutes (Kuwahara et al. Free Radical Biology & Medicine 2010). Each experimental group of muscle-Sod2−/− mice in this study averaged about 7-23 minutes of running time before administration (data not shown), indicating exercise intolerance similar to our previous reports. We have now added a description of the actual running times of wild-type and muscle-Sod2−/− mice to the ‘Results’ section.

‘In our previous report, we demonstrated that wild-type mice can complete a treadmill run for 2 h, whereas muscle-Sod2−/− mice run for less than 10 minutes [4]. Each experimental group of muscle-Sod2−/− mice in this study averaged about 7-23 minutes of running time before administration (data not shown), indicating exercise intolerance similar to our previous reports.’

  1. Lack of biochemical evidence. After the administration, some of the compounds show promising effects in improving muscle functions in muscle-Sod2-/- mouse, any further interpretation on that? For example, as mentioned by the Authors in the introductions, in muscle-Sod2-/- mouse model, the muscle fatigue intolerance could be attributed to impaired mitochondrial function, increased ROS or oxidative stress, damaged or oxidized proteins with elevated oxidative markers. Then if the muscle fatigue intolerance situation was alleviated by the administration of some compounds, was it due to the attenuation of the oxidative stress, or reduced ROS level, or increased mitochondrial function?   

Response: The purpose of this study was to find substances that improve the motor function by in vivo screening in muscle fatigue models. We speculate that the enhancement of the mitochondrial function via ROS scavenging is one mechanism for improving the motor function. However, in this study, we were unable to investigate the details concerning the involved mechanism, so it remains unclear. We also believe obtaining that biochemical evidence, such as that concerning the roles of ROS and mitochondria in improving the motor function, is very important. If these detailed mechanisms could be clarified, the possibility of applying substances found through this research to therapeutic strategies for musculoskeletal disease would be further increased.

However, because the purpose of this study was to screen multiple substances, we had to reuse the mice after the clearance period of the administered substances. Therefore, it was very difficult to collect blood and tissues for biochemical analyses in each experimental group. Furthermore, the recent COVID-19 pandemic has had a serious influence on our research environment. Due to the drastic reduction in research funding, we had to significantly reduce the scale of the experiment. We also needed to reduce the housing space of the mice. In addition, many of the reagents used for the biochemical analysis were out of stock, making it impossible for us to carry out additional experiments. Unfortunately, we were also unable to perform additional experiments due to the seven-day revision period. Therefore, additional data were unfortunately difficult to obtain, and the interpretation of the results obtained in this paper is thus limited.

We have now added a description of the need for biochemical evidence and details concerning the research limitations to the ‘Discussion’ section.

‘In the present study, we performed the trials in a limited number of mice and did not examine biochemical markers in detail, thereby limiting the interpretation of the results obtained.’

‘..., we also need to examine other biochemical markers, such as ROS levels, ATP levels, and the mitochondrial respiratory capacity in detail.’

  1. Vulnerable design of the compounds administration. For the compounds delivery, there is only one time administration to the mice for improving their treadmill performance, I highly doubt the outcome of this “single treatment”. Is there any proof to show the pharmacokinetic of the targeted compound, any evidence to show the levels of the compound in the blood and in the muscle following time?

Response: As suggested, the pharmacological effects on the motor function of substances may be associated with pharmacokinetics after administration. For the majority of this study, we analyzed animals at 24 h after I. P. administration, so the pharmacological action of the substances may have been more enhanced if we had analyzed the peak pharmacokinetics. We have now added references and a description of the association with pharmacokinetics to the ‘Discussion’ section.

‘The pharmacological effects on the motor function of substances may be associated with the pharmacokinetics after administration. In rats, astaxanthin accumulated in skeletal muscle at 24 h after ingestion, suggesting a direct action on skeletal muscle [15]. Human studies also suggest that HMB reaches various organs at nine hours after ingestion [16]. For the majority of this study, we analyzed animals at 24 h after I. P. administration, so the pharmacological action of the substances may have been more enhanced if analyzed at the time point of peak pharmacokinetics. Indeed, despite having antioxidant activity, some compounds failed to improve the motor function in muscle-Sod2-/- mice (Table 1 and 2). These substances have different pharmacokinetics, e.g. differences in the amount reaching muscle tissue and stability, and may have their own appropriate concentrations, analysis timing, and administration methods.’

  1. Vukovich M D. Slater G. Macchi M B. et al. beta-hydroxy-beta-methylbutyrate (HMB) kinetics and the influence of glucose ingestion in humans. J Nutr Biochem 12 (11): 631-639. 2001.
  2. Choi H D. Kang H E. Yang S H. et al. Pharmacokinetics and first-pass metabolism of astaxanthin in rats. Br J Nutr 105 (2): 220-227. 2011.

  1. Lack of rationale on picking up so many compounds. There are so many compounds tested in the study, with most of them having no significant effect, and also there is not enough explanation on why picking up these certain compounds. As a reader, I’d rather see fewer compounds with significant effect, and also with the rationale of why choosing these compounds, and why they would be effective in preserving muscle functions in   muscle-Sod2-/- mouse.

Response: The purpose of this study was to identify substances that improve the motor function by in vivo screening of muscle fatigue models. We selected the substances administered in this study from among available substances, focusing on those with antioxidant, anti-inflammatory, and mitochondrial biosynthetic effects. We have now added a description of our reason for selecting the substances to be administered to the ‘Materials and Methods’ and ‘Results’ sections. As mentioned in our response to Point 3, we have also added a description of the need for biochemical evidence and the research limitations to the ‘Discussion’ section.

‘We selected the substances administered in this study from among the available substances, focusing on those with antioxidant, anti-inflammatory, and mitochondrial biosynthetic effects.’ (Materials and Methods)

‘... with a focus on substances with antioxidant, anti-inflammatory, and mitochondrial biosynthetic effects (Table 1 and 2).’ (Results)

  1. The data presented in this study is only the relative change of the running time with arbitrary unit. I have no clue about their exact running time, and the actual running distance. I think the absolute running time and distance would make more sense to the readers, and also will be more comparable to studies by other peers, at least for the running exercise data in wildtype mouse. 

Response: As noted in our response to Point 2, we have now added a description of the actual running times of wild-type and muscle-Sod2−/− mice to the ‘Results’ section.

Reviewer 2 Report

NATURAL COMPOUNDS THAT ENHANCE THE MOTOR FUNCTION IN A MOUSE MODEL OF MUSCLE FATIGUE

General Commentary

This article presents a very interesting and pertinent question of research of the investigate the improvement effects of polyphenols, vitamins, amino acids, and other chemicals by an in vivo screening of the forced running ability of muscle- Sod2-/- mice.

Congratulations of the excellent manuscript

However, some questions need to be clarified in order to better understand and apply the results found.

MAJOR CONSIDERATION

ABSTRACT

I did not find the objective clearly described in the summary, please describe it clearly.

METHODS

Experimental Design

I suggest the authors insert a first subchapter (Experimental Design) and a figure (with a timeline) in the experimental design of the study, this will make it easier for the reader to understand what was done.

Sample Calculation

I suggest inserting the sample calculation and the sample size of animals used.

DISCUSSION

Limitations

I suggest that the authors address possible limitations of the study, such as the sample size.

Practical Application or Clinical Application

From the results of this study, what are the practical or clinical applications? please write in text or topics. I suggest the authors describe a final subchapter of the discussion (practical or clinical application).

Author Response

Response to Reviewer 2

This article presents a very interesting and pertinent question of research of the investigate the improvement effects of polyphenols, vitamins, amino acids, and other chemicals by an in vivo screening of the forced running ability of muscle- Sod2-/- mice.

Congratulations of the excellent manuscript

However, some questions need to be clarified in order to better understand and apply the results found.

ABSTRACT

I did not find the objective clearly described in the summary, please describe it clearly.

Response: The purpose of this study was to develop a therapeutic strategy for musculoskeletal disease by identifying substances that enhance the motor function from among a number of functional compounds using muscle-Sod2−/− as a muscle fatigue model. We have now described the purpose more clearly in the ‘Abstract’ section.

‘In the present study, to develop a therapeutic strategy for musculoskeletal disease, we searched for substances that enhanced the motor function among functional compounds by in vivo screening using muscle-Sod2-/- mice as a muscle fatigue model.’

METHODS

Experimental Design

I suggest the authors insert a first subchapter (Experimental Design) and a figure (with a timeline) in the experimental design of the study, this will make it easier for the reader to understand what was done.

Response: As suggested, we have added ‘Figure 1’ showing the experimental timeline.

Sample Calculation

I suggest inserting the sample calculation and the sample size of animals used.

Response: We set the minimum number of mice in each experimental group to four for statistical analyses. As suggested, we have added a description of the sample size used to the ‘Materials and Methods’ section.

‘The minimum number of mice in each experimental group was set at four for statistical analyses.’

DISCUSSION

Limitations

I suggest that the authors address possible limitations of the study, such as the sample size.

Response: Because the purpose of this study was to screen multiple substances, we had to reuse the mice after the clearance period of the administered substances. Furthermore, the recent COVID-19 pandemic has had a serious influence on our research environment. Due to the drastic reduction in research funding, we had to significantly reduce the scale of the experiment. We also needed to reduce the housing space of the mice. Due to these limitations, the minimum number of mice in each experimental group was set at four to allow statistical analyses to be performed. Our study is thus limited because it was performed with a limited number of mice. We have now mentioned the research limitations in the ‘Discussion’ section.

‘In the present study, we performed the trials in a limited number of mice and did not examine biochemical markers in detail, thereby limiting the interpretation of the results obtained.’

Practical Application or Clinical Application

From the results of this study, what are the practical or clinical applications? please write in text or topics. I suggest the authors describe a final subchapter of the discussion (practical or clinical application).

Response: Utilizing this muscle fatigue model mice will contribute to the development of materials that can be applied to locomotive syndrome and frailty. As suggested, we have added a description of the practical and clinical applications to the final subchapter of the ‘Discussion’ section.

‘Utilizing this mouse model of muscle fatigue will contribute to the development of materials that can be applied to locomotive syndrome and frailty.’

Reviewer 3 Report

Thank you very much for this manuscript concerning an intersting topic.

Yet, I have several comments.

The introduction should be clarified. Indeed, the way you wrote it [lines 27-34], it is not clear. you seem to make a link between ROS in sports practice and the development of diseases and aging. This is false insofar as physical activity is protective of these phenomena. Please reformulate.

Concerning the substances you used, how did you choose them and why these ones in particular ? Please give this information in the method.

How did your choose the dosage of the molecules ? Please specify.

In your introduction you specified several time that you studied molecules from functional food. However, many of your substances were given intraperitoneally. You have to discuss this point in the discussion, explaining why. And it is really a limit that should be discussed at the end in a limitation part.

Author Response

Response to Reviewer 3

Thank you very much for this manuscript concerning an interesting topic.

Yet, I have several comments.

The introduction should be clarified. Indeed, the way you wrote it [lines 27-34], it is not clear. you seem to make a link between ROS in sports practice and the development of diseases and aging. This is false insofar as physical activity is protective of these phenomena. Please reformulate.

Response: As suggested, we have deleted the beginning of the ‘Introduction’ section for clarity.

Concerning the substances you used, how did you choose them and why these ones in particular? Please give this information in the method.

Response: We selected the substances administered in this study from available substances, focusing on those with antioxidant, anti-inflammatory, and mitochondrial biosynthetic effects. We have now mentioned our reason for selecting the substances to be administered in the ‘Materials and Methods’ and ‘Results’ sections.

‘We selected the substances administered in this study from among the available substances, focusing on those with antioxidant, anti-inflammatory, and mitochondrial biosynthetic effects.’ (Materials and Methods)

‘… with a focus on substances with antioxidant, anti-inflammatory, and mitochondrial biosynthetic effects (Table 1 and 2).’ (Results)

How did you choose the dosage of the molecules? Please specify.

Response: We determined the dosage of each substance by referring to previous reports using experimental animals. To assess the safety, we administered each substance to wild-type mice and confirmed that no deaths or abnormalities occurred. We have now added a description of the dosage and safety of the substances to the ‘Materials and Methods’ section.

‘The dosage of each substance was determined with reference to previous reports using experimental animals. To assess the safety, we administered each substance to wild-type mice and confirmed that no deaths or abnormalities occurred.’

In your introduction you specified several time that you studied molecules from functional food. However, many of your substances were given intraperitoneally. You have to discuss this point in the discussion, explaining why. And it is really a limit that should be discussed at the end in a limitation part.

Response: We conducted this study with a focus on screening substances that enhance the motor function from among a wide variety of substances. Therefore, many substances were administered via intraperitoneal injection, which is a simple procedure and can be performed with small amounts of expensive substance. These substances might have had different pharmacokinetics, e.g. differences in the amount reaching muscle tissue and stability, and may have their own appropriate concentrations, analysis timing, and administration methods. We have now mentioned our reason for selecting this administration method and the experimental limitations to the ‘Materials and Methods’ and ‘Discussion’ sections, respectively.

‘Many substances were administered by intraperitoneal (I. P.) injection, which is a simple procedure that can be performed with small amounts of expensive substances.’ (Materials and Methods)

‘These substances have different pharmacokinetics, e.g. differences in the amount reaching muscle tissue and stability, and may have their own appropriate concentrations, analysis timing, and administration methods.’ (Discussion)

‘In the present study, we performed the trials in a limited number of mice and did not examine biochemical markers in detail, thereby limiting the interpretation of the results obtained.’ (Discussion)

Round 2

Reviewer 1 Report

Thanks for the revisions made by the Authors. I think the responses have answered all of my concerns.

Thanks 

Reviewer 3 Report

Thank you for your clarification.